# Comparative Study on Constitutive Models for 21-4N Heat Resistant Steel during High Temperature Deformation

**DOI:** 10.3390/ma12121893

**Published:** 2019-06-12

**Authors:** Yiming Li, Hongchao Ji, Zhongman Cai, Xuefeng Tang, Yaogang Li, Jinping Liu

**Affiliations:** 1College of Mechanical Engineering, North China University of Science and Technology, Tangshan 063210, China; jxlym666@163.com (Y.L.); czmncst@163.com (Z.C.); jxlyg@ncst.edu.cn (Y.L.); 2National Center for Materials Service Safety, University of Science and Technology Beijing, Beijing 100083, China; 3School of Mechanical Engineering, University of Science and Technology Beijing, Beijing 100083, China; liujp@ustb.edu.cn; 4Department of Mechanical Engineering, The Hong Kong Polytechnic University, Hung Hom, Kowloon, Hong Kong, China

**Keywords:** 21-4N, high temperature deformation, constitutive models, numerical simulation

## Abstract

The Gleeble-1500D thermal simulation test machine was used to conduct the isothermal compression test on 21-4N at the strain rate ($\dot{\varepsilon }$) of 0.01–10 s^−1^, the deformation temperature (*T*) of 1273–1453 K and the maximum deformation is 0.916. The data of the stress-strain (*σ*-*ε*) were obtained. Based on the *σ*-*ε* data, the Johnson-Cook (J-C), modified J-C, Arrhenius and Back-Propagation Artificial Neural Network (BP-ANN) models were established. The accuracy of four models were verified, analyzed and compared. The results show that J-C model has a higher accuracy only under reference deformation conditions. When the deformation condition changes greatly, the accuracy of J-C model is significantly reduced. The coupling effect of *T* and $\dot{\varepsilon }$ of modified J-C model is considered, and the prediction accuracy is greatly improved The Arrhenius model introduces Zener-Hollomon (Z) to represent the coupling effect of *T* and $\dot{\varepsilon }$, it has a fairly high prediction accuracy. And it can predict flow stress (*σ*) accurately at different conditions. The accuracy of BP-ANN model is the highest, but its learning rate is low, the learning and memory are unstable. It has no memory for the weights and thresholds of the completed training. So, there are certain limitations of it in use. Finally, a Finite Element Method (FEM) of the isothermal compression experiment for four models were established, and the distribution of the equivalent stress field, equivalent strain field and temperature field with the deformation degree of 60% were obtained.

## 1. Introduction

21-4N is an austenitic heat-resistant steel with good mechanical properties at high temperature. It can be used in various extreme environments and widely used in the manufacture of various valves [1,2,3]. Its hot deformation behavior is an extremely complex dynamic process accompanied by changes in microstructure, including work hardening, grain growth, dynamic recrystallization and so on [4,5,6]. The flow stress is one of the important indexes for evaluating the deformation ability of metals. The constitutive model is the key to the relationship between reaction the rheological behavior and parameters of metals and alloys. It is an important prerequisite to establish high-precision constitutive model for simulating the deformation process of metals and alloys by finite element method, and also an important means to optimize production process and achieve high-efficiency and high-quality production [7,8,9,10].

In the past few decades, many scholars at home and abroad have done a lot of studies on the constitutive models of different materials. The J-C model is the first used to predict the *σ* of metals due to its simple form, small amount of calculation and significant reduction in the number of experiments [11,12]. Sahu et al. [13] established the J-C model of AA1100 aluminum alloy and simulated and analyzed it by finite element software, the results show that the prediction results have a better accuracy. Limbadri et al. [14,15] predicted the effect of deformation conditions on the stress by using J-C model. Buzyurkin et al. [16] calculated the parameters of J-C model of VT6, OT4 and OT4-0 titanium alloys, and established J-C model to calculate stress. Jia et al. [17] predicted the *σ*-*ε* curves of 7A52 aluminum alloy by using J-C model, which proves that the model can meet the accuracy requirements only under the reference deformation conditions. Wang et al. [18] predicted the rheological behavior of Ti-6Al-4V by J-C model. The finite element simulation was used to verify the accuracy. However, J-C model has limitations because it ignores the mutual coupling between *T* and $\dot{\varepsilon }$, so that it has better accuracy only under reference deformation conditions. In this regard, some scholars have improved J-C model. Grzesik et al. [19] established the J-C model of Inconel 718 nickel-based alloy, and conducted finite element simulation of turning and milling process of jet engine parts with MATLAB software, and obtained better prediction results. Tuğrul et al. [20] proposed to use the evolutionary algorithm to determine the constitutive model parameters of metal materials under high strain rate cutting conditions, and compared with other methods, proved its excellent performance. Zhao et al. [21] established J-C model and modified J-C model of FeCr alloy, and compared the accuracy of them. Zhang et al. [22] predicted the hot rheological behavior of 7075-T6 aluminum alloy by using modified J-C model, and a higher prediction accuracy was obtained. Tan et al. [23] observed the mechanical behavior of 7050-T7451 aluminum alloy by using J-C model and modified J-C model. The stress of 7050-T7451 aluminum alloy was predicted by two models. The prediction results show that the accuracy of modified J-C model has been significantly improved. Wang et al. [24] studied the dynamic behavior of Inconel718, established modified J-C model with *σ*-*ε* data, and verified the prediction accuracy. Bobbili et al. [25] studied the modified J-C model, modified Khan-Huang-Liang (KHL) model and artificial neural network (ANN) model of Ti-13Nb-13Zr alloy respectively. Prawoto et al. [26] studied the bi-ferritic martensite structure of two kinds of hypoeutectoid steel with different carbon content and alloyed elements, and proposed the failure criterion of J-C model to change the ferrite content. Ducobu et al. [27] collected the parameters of the J-C model of Ti6Al4V under different deformation conditions from different literatures, and compared the prediction ability of the J-C model under different parameters, and obtained three parameter sets with better prediction performance. Ranc et al. [28] used the J-C model to analyze the influence of thermal softening on energy. Shrot et al. [29] proposed the Levenberg-Marquardt search algorithm to calculate parameter S of the J-C model, established the J-C model with a specific parameter, and used the model to predict the rheological behavior of materials. He et al. [30] integrated the coupling effect between *T* and $\dot{\varepsilon }$, established modified J-C model of 10%Cr steel, and proved that modified J-C model has a better predictive ability. Although modified J-C model has been improved significantly in accuracy compared to J-C model, it is also limited by deformation conditions. Its prediction accuracy will gradually decrease as the difference that between deformation condition and reference condition increases. The Arrhenius model compensates for this defect by introducing the Z parameter. No matter how the deformation condition changes, its prediction accuracy can achieve the desired effect. Chen et al. [31] studied the hot mechanical behavior of Al-12Zn-2.4Mg-1.2Cu alloy, established Arrhenius model by using the *σ*-*ε* data. And used the average relative error (AARE) and correlation coefficient (R) to evaluate the accuracy, which proves that Arrhenius model has an accurate predictive ability. Abbasi-Bani et al. [32] studied the high temperature deformation behavior of Mg-6Al-1Zn, and established J-C model and Arrhenius model. Li et al. [33] established Arrhenius model of V-5Cr-5Ti alloy by regression analysis, and used it to calculate and analyze the stress. And a better prediction result was obtained. Quan et al. [34,35] studied the hot mechanical behavior of 42CrMo and Ti-6Al-2Zr-1Mo-1V, and established Arrhenius models of two materials to describe hot rheological behavior, obtained accurate prediction results. Samantaray et al. [36] used J-C model and Arrhenius model with strain compensation to characterize the high temperature mechanical properties of 9Cr-1Mo alloy. The results show that J-C model has limitations and cannot well describe the high temperature mechanical behavior of 9Cr-1Mo alloy. The Arrhenius model can accurately predict the stress. Li et al. [37] established Z-A model and Arrhenius model of 7050 aluminum, and the accuracy of the two models were compared by AARE and R. Yan et al. [38] predicted the residual stress distribution of Ti-6Al-4V by using Arrhenius model, and obtained accurate predicted results. Li et al. [39] established Z-A model, J-C model, and Arrhenius model of 28CrMnMoV alloy. The results show that the Arrhenius model can show high prediction accuracy under the whole deformation condition. Compared with Arrhenius model, the accuracy of BP-ANN model is higher. It is one of the widely used ANN models. Yan et al. [40] studied the hot deformation behavior of Al-6.2Zn-0.70Mg-0.30Mn-0.17Zr alloy at 623–773 K and 0.01–20 s^−1^, established its Arrhenius model and BP-ANN model. and compared the accuracy of two models. Han et al. [41] studied the constitutive relation of as-cast 904L, compensated the *σ*-*ε* data with friction factor and temperature, established Arrhenius model and BP-ANN model of it, and compared the accuracy of models. Peng et al. [42] calculated AARE and R of Arrhenius model and BP-ANN model of the as-cast Ti60, compared the accuracy of two models. Vignesh et al. [43] established BP-ANN model of AA1100 aluminum alloy and obtained 99% prediction accuracy. At present, there are few studies on the constitutive relation of 21-4N. As the preferred material for engine valves, it is necessary to conduct a comprehensive study on the constitutive relation.

In this paper, based on the hot deformation behavior of 21-4N, the parameters of four models were calculated by using *σ*-*ε* data. The J-C, modified J-C, Arrhenius and BP-ANN model of 21-4N were established respectively, and the accuracy of four models were verified. And then, the distribution of the equivalent stress field, equivalent strain field and temperature field with the deformation degree of 60% is obtained by using four models for finite element simulation. The results can provide theoretical basis to predict the rheological behavior of 21-4N.

## 2. Experiment

In this study, 21-4N (5Cr21Mn9Ni4N) was selected as the material, the composition of which is as follows: Cr: 20–22%, Mn: 8–10%, Ni: 3.25–4.5%, Si: ≤0.35%, C: 0.48–0.58%, N: 0.35–0.5%, P: ≤0.04%, S: ≤0.03%, Bal: Fe. The test was carried out on Gleeble-1500D high-temperature simulation test machine (Duffer, Durham, NC, USA). The deformation temperature was measured by thermocouple, and the stress-strain data were collected through the computer control system. The size of the sample is *ϕ8 mm × 15 mm*. *T* were set at 1273, 1333, 1393, 1453 K, $\dot{\varepsilon }$ were set at 0.01, 0.1, 1, 10 s^−1^, the maximum deformation was 0.916. Figure 1 shows the heating process of the test. The sample was heated to the target temperature at 10 K/s and holding for 3 min. Finally, the sample was compressed at high temperature with different *ε* and $\dot{\varepsilon }$. After test, water quenching immediately [44,45].

## 3. Result and Discussion

### 3.1. J-C Model

In the 1980s, J-C model was proposed by Johnson and Cook to describe the *σ*-*ε* relationship of metals and alloys under high *T*, high $\dot{\varepsilon }$ and large *ε* [46,47]. When applied, J-C model assumes materials stress conforms to yield criterion and isotropic strain hardening criterion, and its expression is shown as Equation (1):(1)$$\left\{\begin{array}{l} \sigma =\left(A+B\varepsilon^{n}\right)\left(1+C\ln{\dot{\varepsilon }}^{*}\right)\left(1-T^{{*}^{m}}\right)\\ {\dot{\varepsilon }}^{*}=\frac{\dot{\varepsilon }}{{\dot{\varepsilon }}_{0}}\\ T^{*}=\frac{T-T_{r}}{T_{m}-T_{r}} \end{array}\right\}$$

In the equation: *A*—The yield stress, MPa; *n*—Strain hardening index; *C*—Strain rate sensitivity coefficient; ${\dot{\varepsilon }}^{*}$—Dimensionless plastic strain rate; *T**—Dimensionless temperature; *m*—Temperature sensitivity coefficient; ${\dot{\varepsilon }}_{0}$—Reference strain rate, s^−1^; *T_r_*—Reference temperature, K; *B*—Strain hardening parameter, MPa; *T_m_*—Melting temperature, K. Therefore, J-C model can be converted as:(2)$$\sigma =\left(A+B\varepsilon^{n}\right)\left[1+C\ln\left(\frac{\dot{\varepsilon }}{{\dot{\varepsilon }}_{0}}\right)\right]\left[1-{\left(\frac{T-T_{r}}{T_{m}-T_{r}}\right)}^{m}\right]$$

From Equation (2), the J-C model contents three parts [48,49]: strain hardening effect part—$\left(A+B\varepsilon^{n}\right)$, strain rate enhancement effect part—$\left[1+C\ln\left(\frac{\dot{\varepsilon }}{{\dot{\varepsilon }}_{0}}\right)\right]$ and temperature effect part—$\left[1-{\left(\frac{T-T_{r}}{T_{m}-T_{r}}\right)}^{m}\right]$; a total of five material parameters: *A, B, n, C* and *m*.

The J-C model of 21-4N with *ε* of 0.1, 0.15, 0.2, 0.25, 0.3, 0.35, 0.4, 0.45, 0.5, 0.55, 0.6, 0.65, 0.7, 0.75, 0.8, *T* of 1273, 1333, 1393, 1453 K and $\dot{\varepsilon }$ of 0.01, 0.1, 1, 10 s^−1^ was studied. The *T_r_* in this study was set as 1273 K, and the ${\dot{\varepsilon }}_{0}$ was set as 0.01 s^−1^. Under this condition, the yield stress of 21-4N is 107 MPa (*A* = 107 MPa). The melting temperature of 21-4N was set as 1727 K (*T_m_* = 1727 K).

#### 3.1.1. Determination of Parameters n and B

When *T* = *T_r_* = 1273 K and $\dot{\varepsilon }={\dot{\varepsilon }}_{0}=0.01\;s^{-1}$, the $1+C\ln\left(\frac{\dot{\varepsilon }}{{\dot{\varepsilon }}_{0}}\right)$ and $1-{\left(\frac{T-T_{r}}{T_{m}-T_{r}}\right)}^{m}$ in Equation (2) are 1. Therefore, the Equation (2) can be written as:(3)$$\sigma =A+B\varepsilon^{n}$$

Converting and taking the logarithm of Equation (3), and Equation (4) can be obtained:(4)ln(σ−A)=lnB+nlnε

From Equation (4), ln(*σ* − *A*) is linear with ln*ε*. Figure 2 shows the linear relationship of ln(*σ* − *A*) and ln*ε* under different *ε*. The slope is *n*, and the intercept is ln*B*. By calculation, *n* and *B* were obtained: *n* = −0.3642, *B* = 37.3238 MPa.

Taking *A, n* and *B* into Equation (3):(5)$$\sigma =107+37.3238\varepsilon^{-0.3642}$$

#### 3.1.2. Determination of Parameter C

When *T* = *T_r_* = 1273 K, the $1-{\left(\frac{T-T_{r}}{T_{m}-T_{r}}\right)}^{m}$ is 1. That is, the effect of the temperature effect part on the *σ* is neglected. Therefore, the Equation (2) can be converted as Equation (6):(6)$$\sigma =\left(A+B\varepsilon^{n}\right)\left[1+C\ln\left(\frac{\dot{\varepsilon }}{{\dot{\varepsilon }}_{0}}\right)\right]$$

Converting the Equation (6):(7)$$\frac{\sigma }{A+B\varepsilon^{n}}=1+C\ln\left(\frac{\dot{\varepsilon }}{{\dot{\varepsilon }}_{0}}\right)$$

From Equation (7), $\frac{\sigma }{A+B\varepsilon^{n}}$ is linear with $\ln\left(\frac{\dot{\varepsilon }}{{\dot{\varepsilon }}_{0}}\right)$, the slope is *C* and the intercept is 1. Figure 3 shows the linear relationship of $\frac{\sigma }{A+B\varepsilon^{n}}$ and $\ln\left(\frac{\dot{\varepsilon }}{{\dot{\varepsilon }}_{0}}\right)$ under different *ε* and $\dot{\varepsilon }$. By calculation, *C* was obtained: *C* = 0.1449.

#### 3.1.3. Determination of Parameter m

When $\dot{\varepsilon }={\dot{\varepsilon }}_{0}=0.01\;s^{-1}$, the Equation (2) can be converted as Equation (8):(8)$$\sigma =\left(A+B\varepsilon^{n}\right)\left[1-{\left(\frac{T-T_{r}}{T_{m}-T_{r}}\right)}^{m}\right]$$

Converting the Equation (8):(9)$$\ln\left(1-\frac{\sigma }{A+B\varepsilon^{n}}\right)=m\ln\left(\frac{T-T_{r}}{T_{m}-T_{r}}\right)$$

From Equation (9), $\ln\left(1-\frac{\sigma }{A+B\varepsilon^{n}}\right)$ is linear with $\ln\left(\frac{T-T_{r}}{T_{m}-T_{r}}\right)$, the slope is m. Figure 4 shows the linear relationship of $\ln\left(1-\frac{\sigma }{A+B\varepsilon^{n}}\right)$ and $\ln\left(\frac{T-T_{r}}{T_{m}-T_{r}}\right)$ under different *ε* and *T*. By calculation, *m* was obtained: *m* = 0.4223.

The J-C model of 21-4N is obtained by substituting the above calculated parameters into Equation (2):(10)$$\begin{array}{ll} \sigma = & \left(107+37.3238\varepsilon^{-0.3642}\right)\\ & \left(1+0.1449\ln\frac{\dot{\varepsilon }}{0.01}\right)\left[1-{\left(\frac{T-1273}{454}\right)}^{0.4223}\right] \end{array}$$

### 3.2. Modified J-C Model

Obviously, J-C model is established under the assumption that strain (*ε*), strain rate ($\dot{\varepsilon }$) and deformation temperature (*T*) do not affect each other. However, it is found that J-C model has a higher accuracy only under the reference deformation condition by comparing the stresses which were obtained by the Equation (10) with the experimental stresses (Figure 11). When the deformation condition is more and more different from the reference deformation condition, the prediction accuracy will be lower and lower. For the purpose of improving the accuracy of J-C model, Lin, Mirza, Li, Hou et al. [50,51,52,53,54,55] studied and modified J-C model. By comparison, it is found that the values of σ were obtained by the one-dimensional quadratic or cubic equation are closer to the experimental values than the values obtained by the power function. Therefore, the strain hardening effect part of J-C model was replaced, and other two parts were also adjusted accordingly, as shown in Equation (11):(11)$$\sigma =\left(A_{1}+B_{1}\varepsilon +B_{2}\varepsilon^{2}+B_{3}\varepsilon^{3}\right)\left(1+C_{1}\ln{\dot{\varepsilon }}^{*}\right)\exp\left[\left(\lambda_{1}+\lambda_{2}\ln{\dot{\varepsilon }}^{*}\right)\left(T-T_{r}\right)\right]$$

In the equation: *A*_1_, *B*_1_, *B*_2_, *B*_3_, *C*_1_, *λ*_1_, *λ*_2_ are all parameters related to the material, and other physical quantities have the same meaning as in J-C model.

Similar to the solution process for solving the J-C model, the *T_r_* is 1273 K, the ${\dot{\varepsilon }}_{0}$ is 0.01 s^−1^, and the *T_m_* is 1727 K.

#### 3.2.1. Determination of Parameters A_1_, B_1_, B_2_, B_3_

When *T* = *T_r_* = 1273 K and $\dot{\varepsilon }={\dot{\varepsilon }}_{0}=0.01\;s^{-1}$, the Equation (11) can be converted as Equation (12):(12)$$\sigma =A_{1}+B_{1}\varepsilon +B_{2}\varepsilon^{2}+B_{3}\varepsilon^{3}$$

The *σ* under the condition of *ε* range of 0.05–0.8 are taken as the research object, and the coefficients in Equation (12) are solved. Figure 5 shows the *σ*-*ε* data in the *ε* range of 0.05–0.8 and the curve fitted by Equation (12). The values of *A_1_*, *B*_1_, *B*_2_ and *B*_3_ in Equation (12) can be obtained by calculation (As shown in Table 1).

#### 3.2.2. Determination of Parameter C_1_

When *T* = *T_r_* = 1273 K, the Equation (11) can be written as:(13)$$\sigma =\left(A_{1}+B_{1}\varepsilon +B_{2}\varepsilon^{2}+B_{3}\varepsilon^{3}\right)\left(1+C_{1}\ln{\dot{\varepsilon }}^{*}\right)$$

Converting the Equation (13):(14)$$\frac{\sigma }{A_{1}+B_{1}\varepsilon +B_{2}\varepsilon^{2}+B_{3}\varepsilon^{3}}=1+C_{1}\ln\dot{\varepsilon }*$$

From Equation (14), $\frac{\sigma }{A_{1}+B_{1}\varepsilon +B_{2}\varepsilon^{2}+B_{3}\varepsilon^{3}}$ is linear with $\ln{\dot{\varepsilon }}^{*}$, the slope is *C*_1_. Figure 6 shows the linear relationship of $\frac{\sigma }{A_{1}+B_{1}\varepsilon +B_{2}\varepsilon^{2}+B_{3}\varepsilon^{3}}$ and $\ln{\dot{\varepsilon }}^{*}$ under different ε and $\dot{\varepsilon }$. By calculation, *C*_1_ was obtained: *C*_1_ = 0.2002.

#### 3.2.3. Determination of Parameters λ_1_, λ_2_

Converting the Equation (11):(15)$$\frac{\sigma }{\left(A_{1}+B_{1}\varepsilon +B_{2}\varepsilon^{2}+B_{3}\varepsilon^{3}\right)\left(1+C_{1}\ln\dot{\varepsilon }*\right)}=\exp\left[\left(\lambda_{1}+\lambda_{2}\ln{\dot{\varepsilon }}^{*}\right)\left(T-T_{r}\right)\right]$$

Taking the logarithm:(16)$$\ln\frac{\sigma }{\left(A_{1}+B_{1}\varepsilon +B_{2}\varepsilon^{2}+B_{3}\varepsilon^{3}\right)\left(1+C_{1}\ln\dot{\varepsilon }*\right)}=\left(\lambda_{1}+\lambda_{2}\ln{\dot{\varepsilon }}^{*}\right)\left(T-T_{r}\right)$$

Figure 7 shows the linear relationship of $ln\frac{\sigma }{\left(A_{1}+B_{1}\varepsilon +B_{2}\varepsilon^{2}+B_{3}\varepsilon^{3}\right)\left(1+C_{1}ln{\dot{\varepsilon }}^{*}\right)}$ and $T-T_{r}$ under different $\dot{\varepsilon }$. By calculation, $\lambda_{1}+\lambda_{2}\ln{\dot{\varepsilon }}^{*}$ were obtained in different $\dot{\varepsilon }$ (As shown in Table 2).

Let $\lambda_{1}+\lambda_{2}\ln{\dot{\varepsilon }}^{*}$, then λ is linear with $\ln{\dot{\varepsilon }}^{*}$. Where $\lambda_{1}$ is intercept and $\lambda_{2}$ is slope. Figure 8 shows the linear relationship of λ and $\ln{\dot{\varepsilon }}^{*}$, the values of $\lambda_{1}$ and $\lambda_{2}$ can be obtained by calculation: $\lambda_{1}=-0.00647$, $\lambda_{2}=0.00042$.

Taking $A_{1},\;B_{1},\;B_{2},\;C_{1},\;\lambda_{1},\lambda_{2}$ obtained from above calculation into Equation (11) to obtain the modified J-C model of 21-4N:(17)$$\begin{array}{l} \sigma =\left(179.67+113.27\varepsilon -647.79\varepsilon^{2}+465.55\varepsilon^{3}\right)\left(1+0.2002\ln{\dot{\varepsilon }}^{*}\right)\\ \;\;\exp\left[\left(-0.00647+0.00042\ln{\dot{\varepsilon }}^{*}\right)\left(T-T_{r}\right)\right] \end{array}$$

### 3.3. Arrhenius Model

There is a relationship between *σ* and *T*, $\dot{\varepsilon }$, ε in hot deformation of metals and alloys. It can be expressed by Arrhenius model which proposed by Sellers and Tegart [56,57,58,59]:(18)$$\dot{\varepsilon }=f\left(\sigma \right)\exp\left(-Q/RT\right)$$

Among them:(19)$$f\left(\sigma \right)=\left\{ \begin{array}{ll} A\sigma^{m} & \left(\alpha \sigma <0.8\right)\\ B\exp(\beta \sigma ) & \left(\alpha \sigma >1.2\right)\\ C{\left[\sinh(\alpha \sigma )\right]}^{n} & \left(For\;all\;\sigma \right) \end{array}\right.$$

*α, β, m* meets the following relationship:(20)α=β/m

In the equation: $\dot{\varepsilon }$—Strain rate, s^−1^; *T*—Absolute temperature, K; *R*—Gas parameter, 8.3145 *J/(mol·K)*; *Q*—Deformation activation energy, *kJ/mol*; *A, B, C*—Material parameter; *m, n*—Stress index; *α*, *β*—Stress level parameter.

There is a certain functional relationship between $\dot{\varepsilon }$, *Q* and *T* during hot deformation. Z parameters can be introduced to express this relationship [60,61,62,63]:(21)$$Z=\dot{\varepsilon }\exp\left(-Q/RT\right)$$

Taking the flow stress function under full pressure in Equation (19) into Equation (18):(22)$$\dot{\varepsilon }=C{\left[\sinh(\alpha \sigma )\right]}^{n}\exp\left(-Q/RT\right)$$

Combining Equation (21) with Equation (22), the Arrhenius constitutive equation can be described with Z parameters. As shown in Equation (23):(23)$$\sigma =\frac{1}{\alpha }\ln\left\{{\left(\frac{Z}{C}\right)}^{1/n}+{\left[{\left(\frac{Z}{C}\right)}^{2/n}+1\right]}^{1/2}\right\}$$

The three expressions in Equation (19) were respectively taken into Equation (18), then taking the logarithm of the equations. The linear relationships between $\ln\dot{\varepsilon }$ and lnσ, σ, ln[sinh(ασ)], ln[sinh(ασ)] and 1/*T*, ln*Z* and ln[sinh(ασ)] were obtained. The values of *m*, *β*, *n*, *n*_1_ and ln*C* were obtained. The values of α and *Q* can be obtained by calculation. The results were shown in Table 3.

Figure 9 shows the relationships between *m, β, α, n, Q*, ln*C* and *ε*.

The curves in the figure were respectively fitted by the sixth-order polynomial form shown by the Equation (24), and the fitting results were shown in Table 4 [64,65]. Taking the data in Table 4 into Equation (24) to obtain function relationships between *m, β, α, n, Q*, ln*C* and *ε*. The values of *m, β, α, n, Q, and* ln*C* under different strains can be calculated by the Equation (24), and the *σ* under different strains can be obtained correspondingly.
(24)$$\left\{\begin{array}{l} m=A_{00}+A_{11}\varepsilon +A_{22}\varepsilon^{2}+A_{33}\varepsilon^{3}+A_{44}\varepsilon^{4}+A_{55}\varepsilon^{5}+A_{66}\varepsilon^{6}\\ \beta =B_{00}+B_{11}\varepsilon +B_{22}\varepsilon^{2}+B_{33}\varepsilon^{3}+B_{44}\varepsilon^{4}+B_{55}\varepsilon^{5}+B_{66}\varepsilon^{6}\\ \alpha =C_{00}+C_{11}\varepsilon +C_{22}\varepsilon^{2}+C_{33}\varepsilon^{3}+C_{44}\varepsilon^{4}+C_{55}\varepsilon^{5}+C_{66}\varepsilon^{6}\\ n=D_{00}+D_{11}\varepsilon +D_{22}\varepsilon^{2}+D_{33}\varepsilon^{3}+D_{44}\varepsilon^{4}+D_{55}\varepsilon^{5}+D_{66}\varepsilon^{6}\\ Q=E_{00}+E_{11}\varepsilon +E_{22}\varepsilon^{2}+E_{33}\varepsilon^{3}+E_{44}\varepsilon^{4}+E_{55}\varepsilon^{5}+E_{66}\varepsilon^{6}\\ \ln C=F_{00}+F_{11}\varepsilon +F_{22}\varepsilon^{2}+F_{33}\varepsilon^{3}+F_{44}\varepsilon^{4}+F_{55}\varepsilon^{5}+F_{66}\varepsilon^{6} \end{array}\right\}$$

### 3.4. BP-ANN Model

The BP model is an artificial network that is calculated and trained according to error back propagation. Its Schematic diagram is shown in Figure 10. The working process includes the information forward propagation and error back propagation. That is, the information is input from the input layer through the transfer function, propagated in the forward direction, and output by the output layer. When the error between the output information and the expected information exceeds the normal range, the error signal will be returned in the original way (That is, error back propagation). And the weight of each layer of neurons will be modified through the training network, and thus repeated until the output information error reaches a reasonable range, the training is completed [66,67,68].

When 21-4N undergoes high temperature degeneration, its constitutive relationship can be expressed as:(25)$$\sigma =\sigma \left(\varepsilon ,\dot{\varepsilon },T\right)$$

From Equation (25), the σ is a function of *ε*, $\dot{\varepsilon }$ and *T*. Therefore, there are three input components and one output component in the BP-ANN model.

Since the components of the input layer have large differences in numerical values, for the purpose of data concentration on one or more neurons which causes the low accuracy of the model. It is necessary to normalize the input data. In general, the Equation (26) is often used to normalize *T* and *σ* [69,70]:(26)$$Y=\frac{X-0.95X_{\min}}{1.05X_{\max}-0.95X_{\min}}$$

In the equation: X—Raw data obtained from the experiment; X_min_, X_max_—The extreme values of X; Y—X normalized vector value.

Equation (27) is often used to normalize the $\dot{\varepsilon }$:(27)$$\dot{\varepsilon }=\frac{3+\ln\dot{\varepsilon }-0.95\left(3+\ln{\dot{\varepsilon }}_{\min}\right)}{1.05\left(3+\ln{\dot{\varepsilon }}_{\max}\right)-0.95\left(3+\ln{\dot{\varepsilon }}_{\min}\right)}$$

In this paper, the MATLAB software was used for programming calculation, and the BP-ANN model was trained by using the normalized data. The training function is TRAINLM, the learning function is LEARNGD. The activation function from input layer to hidden layer is TANSIG, and the returned is PURELIN. The 3 × 10 × 10 × 1 four-layer BP-ANN structure was selected for training, and the error target was set located 10^−3^. Figure 11 shows the training results. After 47 iterations, the training results reach the error target, the established model converges rapidly and the training is completed.

### 3.5. Analysis of Constitutive Equation Accuracy

The comparison of stresses between experimental and predicted of four models are shown in Figure 12, Figure 13, Figure 14 and Figure 15.

From Figure 12, J-C model has a higher accuracy only at *T_r_* and ${\dot{\varepsilon }}_{0}$. The predicted stresses are greatly different from the experimental stresses under other conditions. At low $\dot{\varepsilon }$ and low *T*, J-C model has a higher accuracy than that at high $\dot{\varepsilon }$ and high T. It indicates that J-C model is suitable for predicting σ at low $\dot{\varepsilon }$ and low T. When the deformation temperature is constant, the prediction accuracy of J-C model gradually decreases with the increase of strain rate. When the strain rate reaches 10 s^−1^, the model basically loses its prediction ability. When the strain rate is 10 s^−1^, as the deformation temperature increases, the accuracy of the model becomes lower and lower, and the prediction ability is lost when the deformation temperature is 1333 K. The *T* and $\dot{\varepsilon }$ have an effect on the *σ* and also affect each other. With the increase of *T* and $\dot{\varepsilon }$, this effect is more obviously.

Modified J-C model improves strain hardening effect part, considers the coupling effect between the *T* and $\dot{\varepsilon }$. From Figure 13, the accuracy of modified J-C model is significantly higher than that of J-C model, which indicates that the coupling effect between the influencing factors needs to be considered when predicting σ. Modified J-C model is also not suit to predict σ at high $\dot{\varepsilon }$ [71,72].

Arrhenius model introduces the Z parameter in the calculation process. The Z parameter combines the coupling effect of T and $\dot{\varepsilon }$. Figure 14 shows the stresses obtained by Arrhenius model. The predicted stresses are highly consistent with experimental data, indicating that it has high prediction accuracy at the experimental conditions. It shows that the model has a wide range of application [73,74].

Figure 15 shows BP-ANN model of 21-4N. The predicted data are evenly distributed on the *σ*-*ε* curves, indicating that the BP-ANN model has a high accuracy under the experimental conditions. Moreover, the BP-ANN model has only two hidden layers in this study, and the precision has reached 99.769%, which indicates that when the hidden layers is more than two layers, the accuracy of it will be infinitely close to 100%. The predicted stresses obtained by the BP-ANN model are basically identical to that of the experimental [75,76].

The average relative error (AARE) and correlation coefficient (R) is introduced to verify the accuracy of four models. The values of AARE and R more accurately reflect the linear relationship and effective information between experimental and predicted. Equations (28) and (29) show the expressions of AARE and R:(28)$$AARE(\%)=\frac{1}{N}\sum_{i=1}^{N}\left|\frac{E_{i}-P_{i}}{E_{i}}\right|\times 100\%$$
(29)$$R=\frac{\sum_{i=1}^{n}\left(E_{i}-\bar{E})(P_{i}-\bar{P}\right)}{\sqrt{{\sum_{i=1}^{n}\left(E_{i}-\bar{E}\right)}^{2}{\left(P_{i}-\bar{P}\right)}^{2}}}$$

Figure 16 shows the comparison of stresses between experimental and prediction of four models. These R and AARE of four models are calculated by using Equations (28) and (29). Table 5 shows the results. From Table 5, the accuracy of modified J-C model is significantly higher than J-C model. And the accuracy of the BP-ANN model is the highest.

### 3.6. Finite Element Simulation of Four Models

In order to verify the practicability of the four models, the Deform-3D software was used to simulation the isothermal compression process. Taking the deformation temperature of 1393 K and strain rate of 10 s^−1^ as an example, the four established models were input into Deform-3D software for numerical simulation. Figure 17 shows the 3D mesh geometric model simulated by finite element method and the compression process of the simulated sample. The simulated deformation is consistent with the actual compression test. There is no interruption or error report in the simulation process, which indicates that the four established models are correct and can be input into the simulation software for simulation and the simulation accuracy is acceptable.

Figure 18 shows the equivalent stress field, equivalent strain field and temperature field of the samples simulated by four models. It can be seen from the figure that the equivalent stress field, equivalent strain field and temperature field of the samples obtained by the simulation of the four models are all symmetrical structures, indicating that the samples are under uniform stress in the compression process. The maximum stress values all appear in the place where the sample and the mold contact, indicating that the contact surface between the sample and the mold is the main stress area. The equivalent strain field is divided into three or more regions, indicating that the deformation of the sample is not uniform. The maximum strain appears in the center of the sample, because the center part of the sample is the main deformation area, and it is subject to the common pressure of the upper die and the lower die, and the friction force is smaller than the upper and lower contact surfaces of the sample, so the deformation degree is the maximum. The temperature of the upper and lower surfaces of the sample is the lowest because of the heat transfer between the sample and the mold part. The temperature in the center of the sample is the highest, because in the compression process, the sample will generate a lot of heat due to deformation, and the center part only transfers heat with the surrounding environment, the heat dissipation is far less than the heat transfer between the sample and the mold, so the temperature is the highest.

## 4. Conclusions


The *T* and $\dot{\varepsilon }$ are the main influencing factors of 21-4N during hot deformation, and they have a coupling effect on *σ*.J-C model ignores the coupling effect of the $\dot{\varepsilon }$ and *T*. As a result, when the deformation condition changes greatly, the accuracy decreases obviously.Modified J-C model considers the coupling effect of *T* and $\dot{\varepsilon }$. The parameter compensation is carried out on the basis of J-C model. The prediction accuracy has been greatly improved. When the deformation condition changes greatly, the accuracy of it is still within a reasonable range.Arrhenius model uses the Z parameter to express the coupling effect of *T* and $\dot{\varepsilon }$. The prediction accuracy is higher than modified J-C model. It is suitable for the *σ* prediction under the reasonable deformation conditions.The established BP-ANN model has two hidden layers, which is 3 × 10 × 10 × 1 topology, and the training is completed after 47 iterations. And it has a very high accuracy under the conditions allowed by the deformation conditions.All the four models can be input into the finite element software for compression test simulation, and the simulation results are not much different from the experimental results, indicating that the four models established have certain practicability.


## Figures and Tables

**Figure 1 materials-12-01893-f001:**
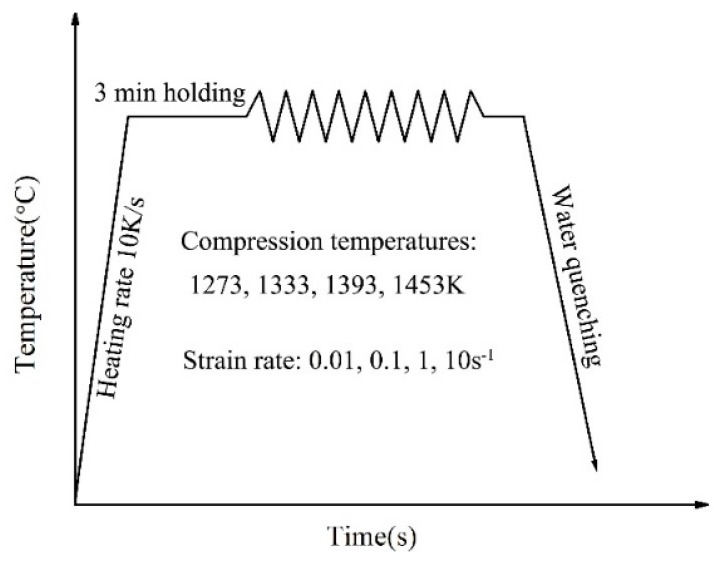
Process flow diagram of the heating process.

**Figure 2 materials-12-01893-f002:**
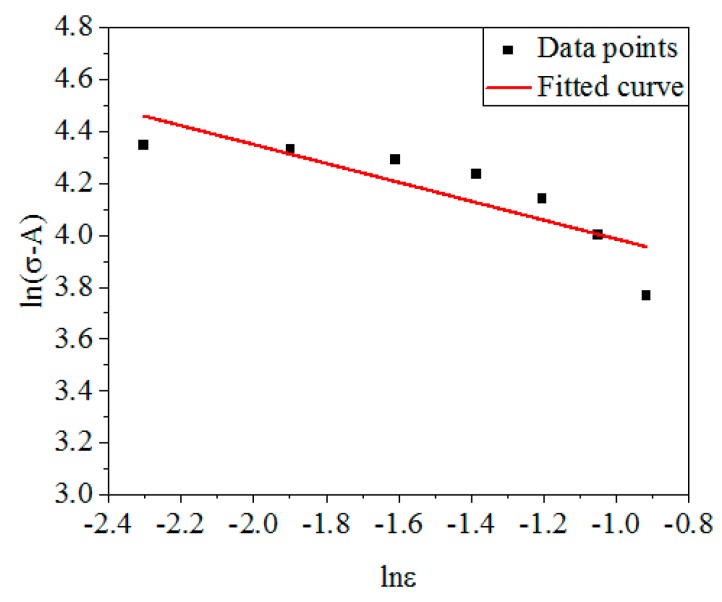
The linear relationship between ln(*σ* − *A*) and ln*ε*.

**Figure 3 materials-12-01893-f003:**
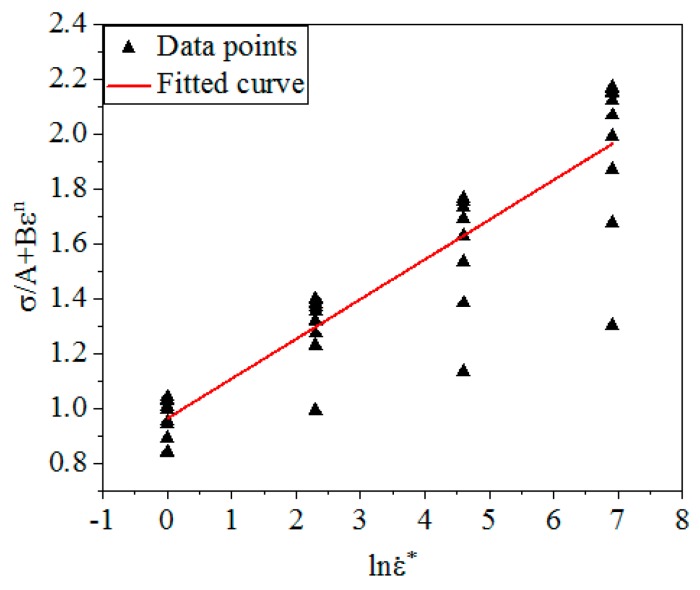
The linear relationship between $\frac{\sigma }{A+B\varepsilon^{n}}$ and $\ln\left(\frac{\dot{\varepsilon }}{{\dot{\varepsilon }}_{0}}\right)$.

**Figure 4 materials-12-01893-f004:**
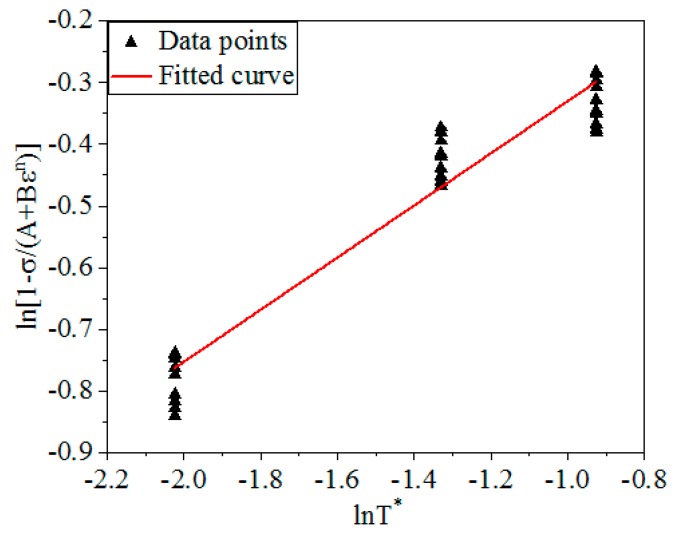
The linear relationship between $\ln\left(1-\frac{\sigma }{A+B\varepsilon^{n}}\right)$ and $\ln\left(\frac{T-T_{r}}{T_{m}-T_{r}}\right)$.

**Figure 5 materials-12-01893-f005:**
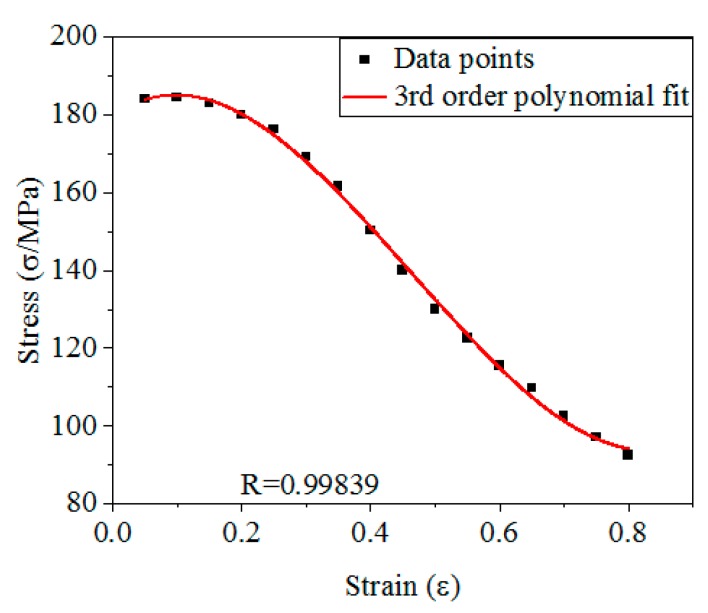
The *σ*-*ε* relationship with *ε* range of 0.05–0.8.

**Figure 6 materials-12-01893-f006:**
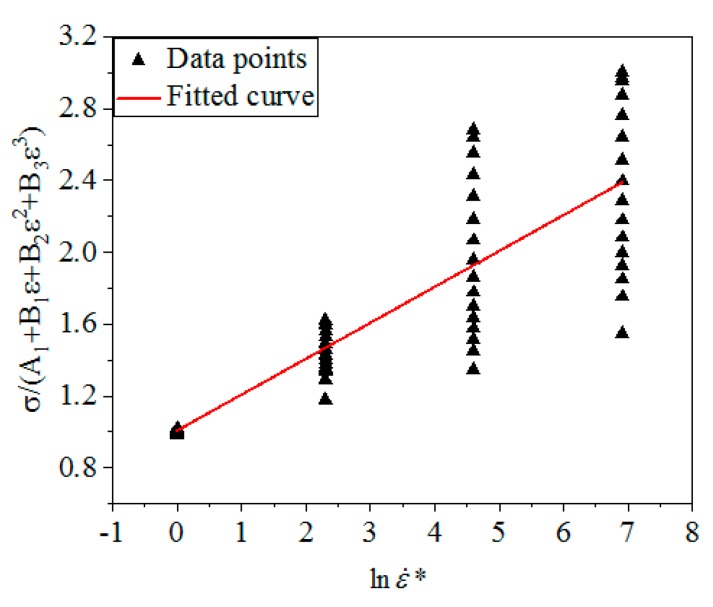
The linear relationship between $\frac{\sigma }{A_{1}+B_{1}\varepsilon +B_{2}\varepsilon^{2}+B_{3}\varepsilon^{3}}$ and $\ln{\dot{\varepsilon }}^{*}$.

**Figure 7 materials-12-01893-f007:**
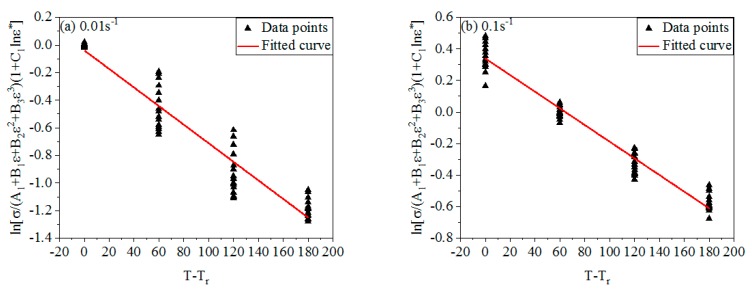
The linear relationship between $\ln\frac{\sigma }{\left(A_{1}+B_{1}\varepsilon +B_{2}\varepsilon^{2}+B_{3}\varepsilon^{3}\right)\left(1+C_{1}\ln{\dot{\varepsilon }}^{*}\right)}$ and $T-T_{r}$ under different $\dot{\varepsilon }$. (**a**) 0.01 s^−1^; (**b**) 0.1 s^−1^; (**c**) 1 s^−1^; (**d**) 10 s^−1^.

**Figure 8 materials-12-01893-f008:**
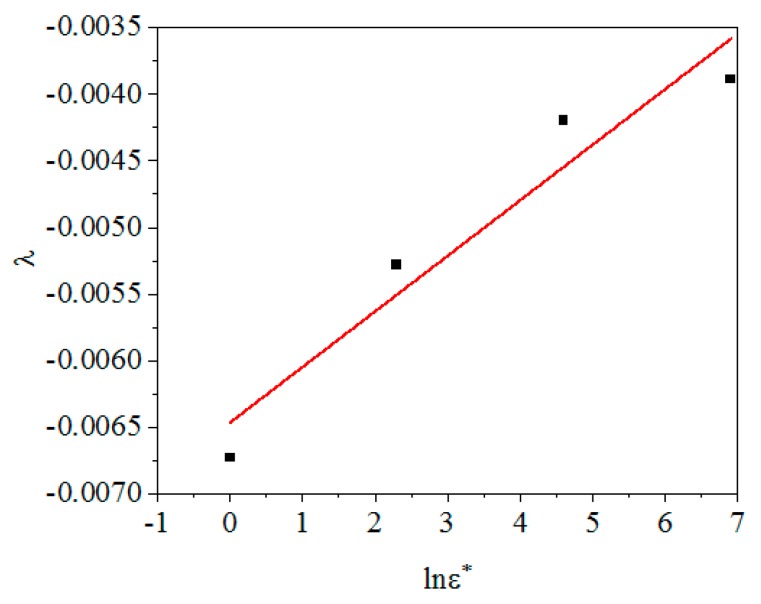
The linear relationship between λ and $\ln{\dot{\varepsilon }}^{*}$.

**Figure 9 materials-12-01893-f009:**
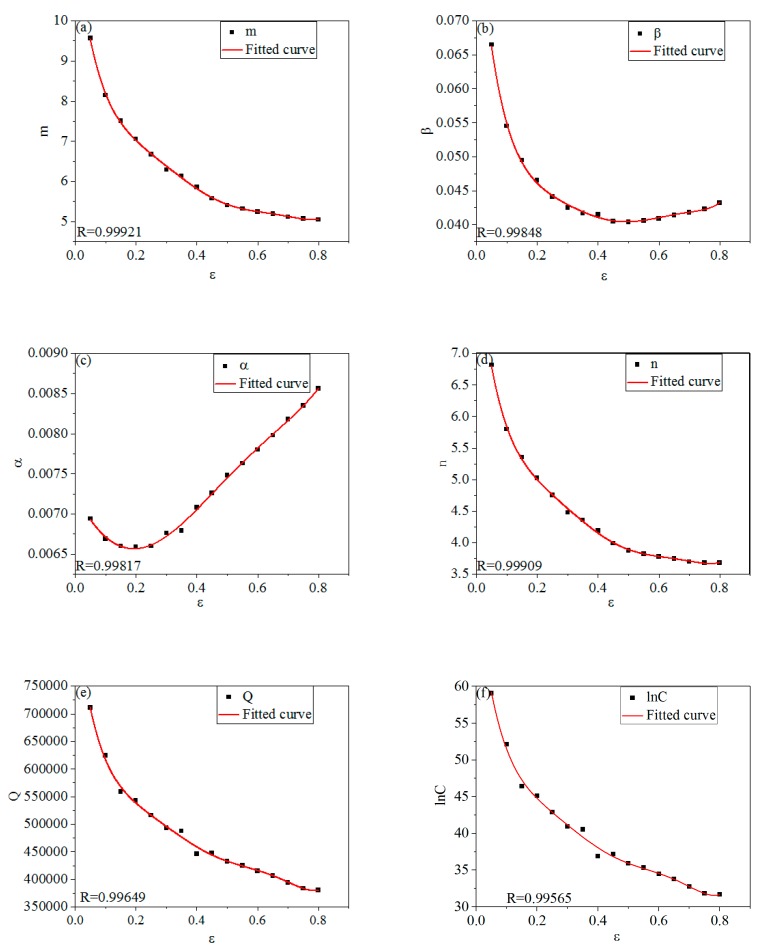
The relationships between *m, β, α, n, Q,* ln*C* and *ε*. (**a**) *m-ε*; (**b**) *β-ε*; (**c**) *α-ε*; (**d**) *n-ε*; (**e**) *Q-ε;* (**f**) ln*C-ε*.

**Figure 10 materials-12-01893-f010:**
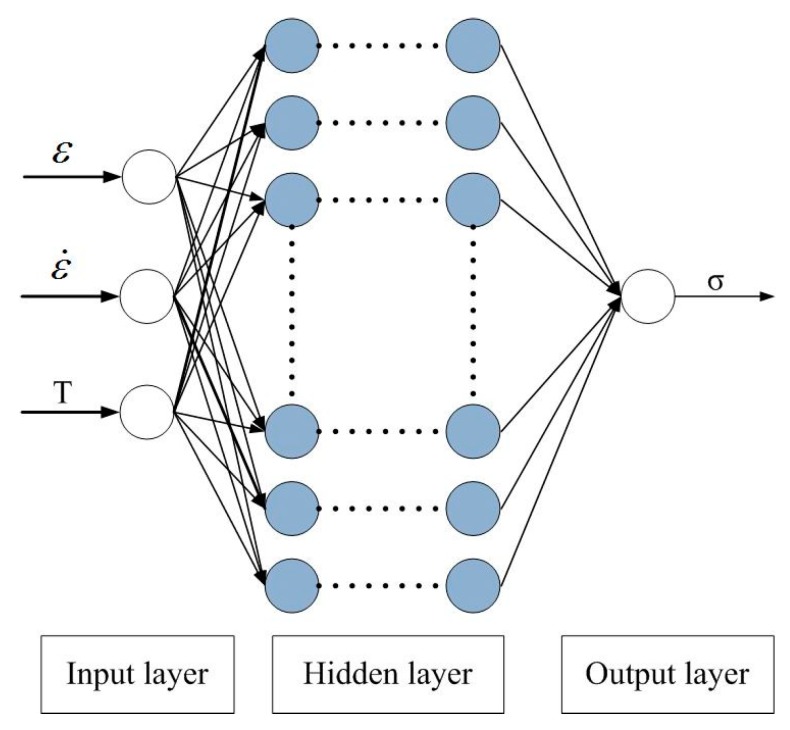
Schematic diagram of artificial neural network model.

**Figure 11 materials-12-01893-f011:**
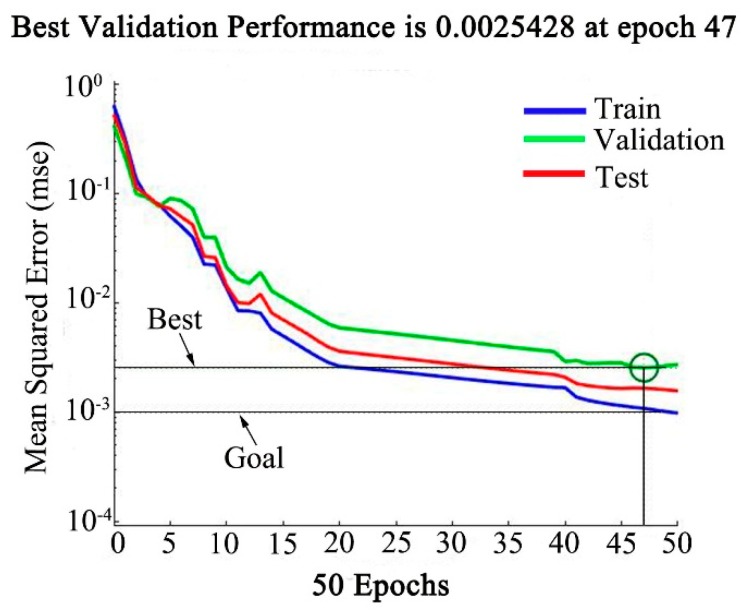
Artificial neural network model training convergence curve.

**Figure 12 materials-12-01893-f012:**
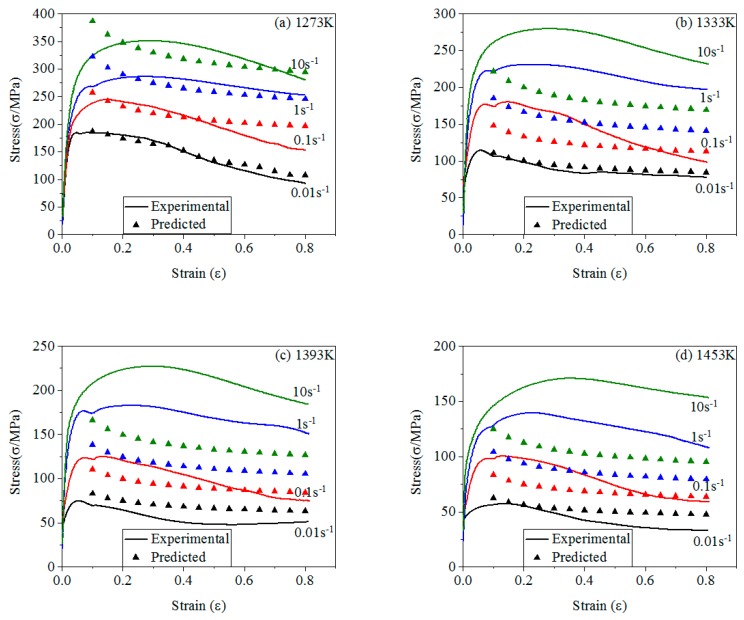
The comparison of stresses between experimental and predicted of J-C model: (**a**) 1273 K; (**b**) 1333 K; (**c**) 1393 K; (**d**) 1453 K.

**Figure 13 materials-12-01893-f013:**
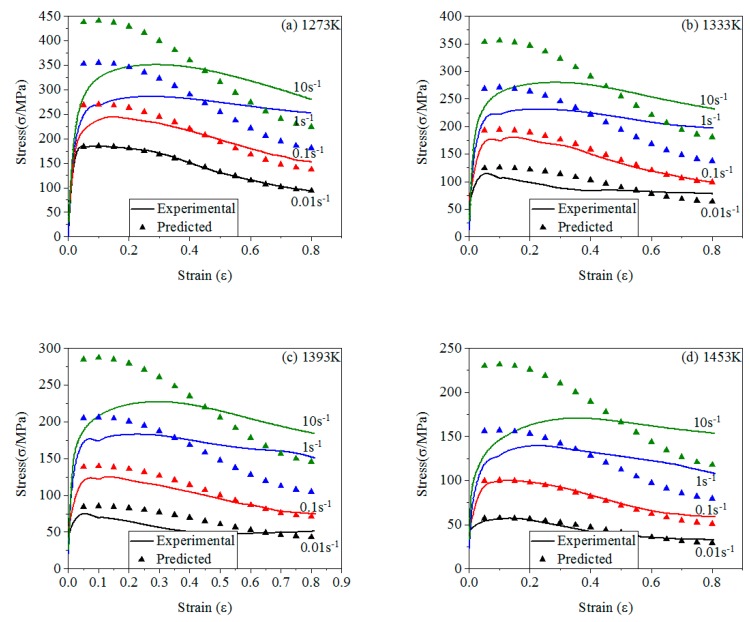
The comparison of stresses between experimental and predicted of modified J-C model: (**a**) 1273 K; (**b**) 1333 K; (**c**) 1393 K; (**d**) 1453 K.

**Figure 14 materials-12-01893-f014:**
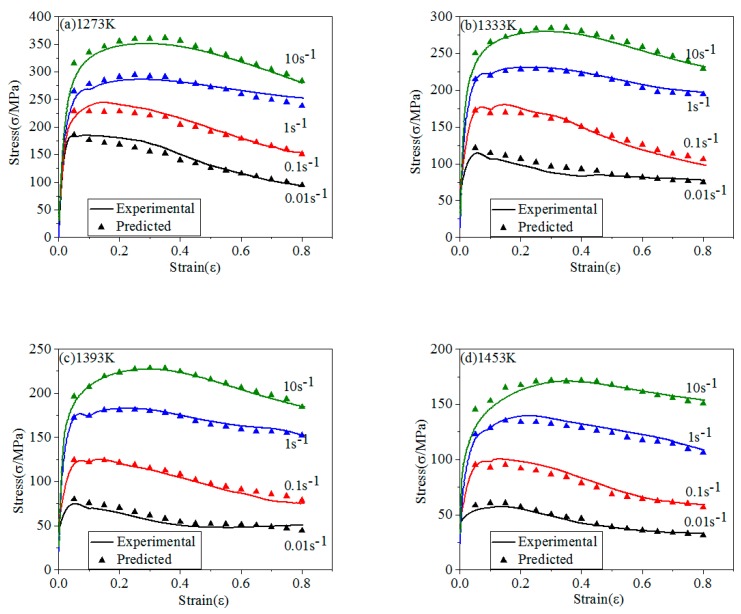
The comparison of stresses between experimental and predicted of Arrhenius model: (**a**) 1273 K; (**b**) 1333 K; (**c**) 1393 K; (**d**) 1453 K.

**Figure 15 materials-12-01893-f015:**
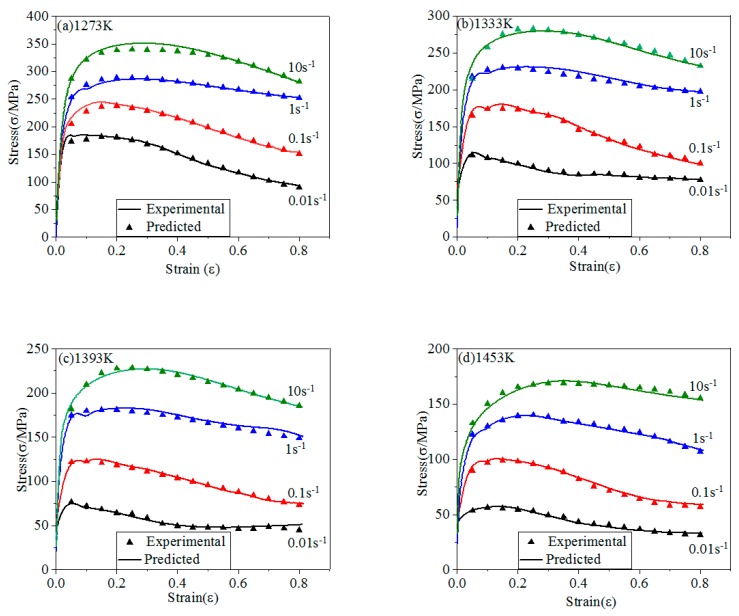
The comparison of stresses between experimental and predicted of BP-ANN model: (**a**) 1273 K; (**b**) 1333 K; (**c**) 1393 K; (**d**) 1453 K.

**Figure 16 materials-12-01893-f016:**
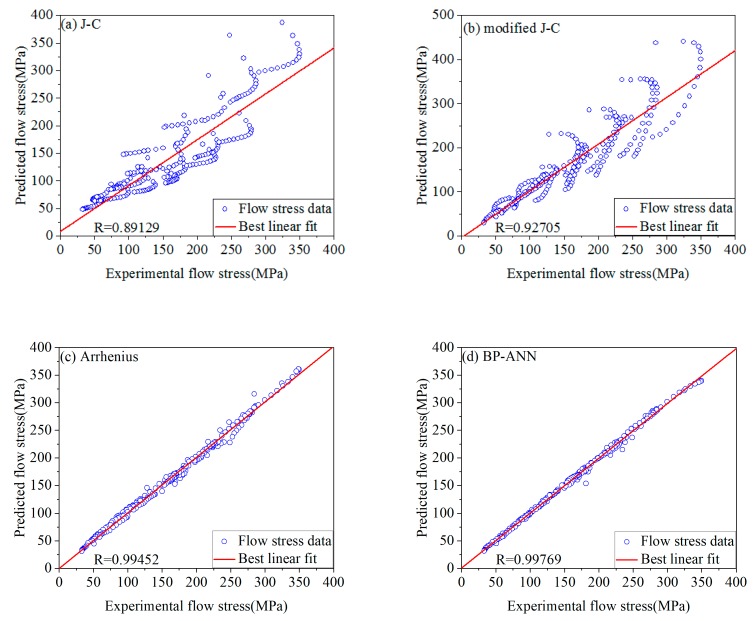
The comparison of experimental and predicted stresses of four models: (**a**) J-C model; (**b**) modified J-C model; (**c**) Arrhenius model; (**d**) BP-ANN model.

**Figure 17 materials-12-01893-f017:**
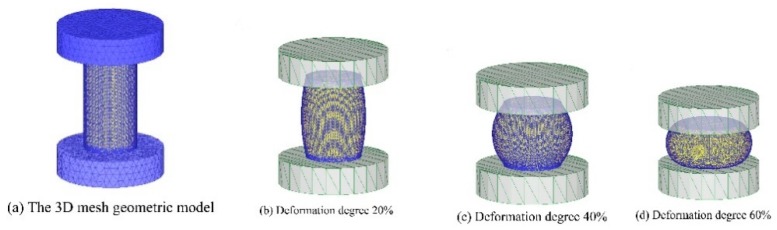
The 3D mesh geometric model and compression process of the simulated sample. (**a**) 3D mesh geometric model; (**b**) Deformation degree 20%; (**c**) Deformation degree 40%; (**d**) Deformation degree 60%.

**Figure 18 materials-12-01893-f018:**
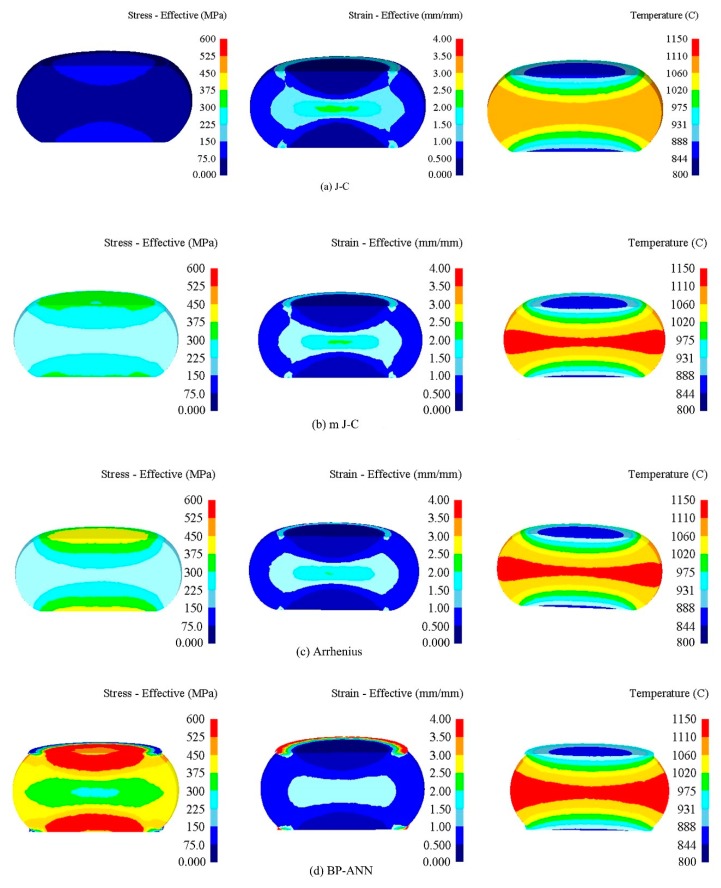
The equivalent stress field, equivalent strain field and temperature field of Four models: (**a**) J-C model; (**b**) modified J-C model; (**c**) Arrhenius model; (**d**) BP-ANN model.

**Table 1 materials-12-01893-t001:** The values of *A*_1_, *B*_1_, and *B*_2_ in Equation (12).

A_1_	B_1_	B_2_	B_3_
179.67 MPa	113.27 MPa	−647.79 MPa	465.55 MPa

**Table 2 materials-12-01893-t002:** The values of $\lambda_{1}+\lambda_{2}\ln{\dot{\varepsilon }}^{*}$ under different $\dot{\varepsilon }$.

$\dot{\varepsilon }$	0.01	0.1	1	10
$\lambda_{1}+\lambda_{2}\ln{\dot{\varepsilon }}^{*}$	−0.00673	−0.00528	−0.00420	−0.00389

**Table 3 materials-12-01893-t003:** Parameter values of 21-4N under different strains.

Parameter	*ε*							
	0.05	0.1	0.15	0.2	0.25	0.3	0.35	0.4
*m*	9.571	8.144	7.505	7.059	6.670	6.288	6.137	5.852
*β*	0.0665	0.0545	0.0495	0.0465	0.0441	0.0425	0.0417	0.0415
*α*	0.00694	0.00669	0.00660	0.00659	0.00660	0.00676	0.00679	0.00708
*n*	6.819	5.797	5.356	5.025	4.752	4.475	4.355	4.188
*n* _1_	12,547.3	12,957.7	12,549.5	12,985.7	13,055.1	13,255.2	13,463.4	12,806.4
*Q*	711,389	624,550	558,860	542,547	515,813	493,191	487,504	445,933
*lnC*	59.087	52.144	46.428	45.129	42.896	40.948	40.548	36.900
Parameter	*ε*							
	0.45	0.5	0.55	0.6	0.65	0.7	0.75	0.8
*m*	5.577	5.404	5.321	5.238	5.187	5.112	5.068	5.046
*β*	0.0405	0.0404	0.0406	0.0409	0.0414	0.0418	0.0423	0.0432
*α*	0.00726	0.00748	0.00763	0.00780	0.00798	0.00818	0.00835	0.00856
*n*	3.988	3.876	3.822	3.775	3.744	3.697	3.677	3.682
*n* _1_	13,504.1	13,429.8	13,384.9	13,221.8	13,063.5	12,831.4	12,541.5	12,421.2
*Q*	447,772	432,802	425,345	414,995	406,660	394,420	383,423	380,262
*lnC*	37.188	35.935	35.350	34.497	33.799	32.752	31.831	31.691

**Table 4 materials-12-01893-t004:** The values of parameter in Equation (24).

*m*	*β*	*α*	*n*	*Q*	lnC
A_00_ = 12.05	B_00_ = 0.087	C_00_ = 0.00725	D_00_ = 8.57	E_00_ = 8.92E5	F_00_ = 73.81
A_11_ = −65.87	B_11_ = −0.550	C_11_ = −0.00737	D_11_ = −46.54	E_11_ = −4.78E6	F_11_ = −388.15
A_22_ = 377.37	B_22_ = 3.092	C2_2_ = 0.01941	D_22_ = 266.35	E_22_ = 2.84E7	F_22_ = 2291.91
A_33_ = −1227.28	B_33_ = −9.605	C_33_ = 0.01259	D_33_ = −870.35	E_33_ = −9.51E7	F_33_ = −7712.90
A_44_ = 2143.97	B_44_ = 16.401	C_44_ = −0.06592	D_44_ = 1533.55	E_44_ = 1.72E8	F_44_ = 14044.30
A_55_ = −1882.28	B_55_ = −14.312	C_55_ = 0.05639	D_55_ = −1359.92	E_55_ = −1.57E8	F_55_ = −12928.66
A_66_ = 652.95	B_66_ = 4.985	C_66_ = −0.01197	D_66_ = 476.78	E_66_ = 5.68E7	F_66_ = 4707.99

**Table 5 materials-12-01893-t005:** R and AARE of the four models.

Types of models	J-C	modified J-C	Arrhenius	BP-ANN
AARE	19.4704%	13.7428%	3.3774%	1.7634%
R	0.83255	0.92705	0.99452	0.99769

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
