# Peer review of "Comparative Study on Constitutive Models for 21-4N Heat Resistant Steel during High Temperature Deformation"

_materials, 2019, doi:10.3390/ma12121893_

Reviewer 1 Report

The paper presents interesting results and calculation data with respect to the high temperature deformation behavior of a heat resistant steel. It is admirable that the authors compared four different models considering strain rate- and temperature-dependent mechanical behavior. The results presented may be useful to the relevant community. I however believe that there are a few parts where the authors can improve the technical quality of the manuscript. I would like to comment on some points in the manuscript:

2. Experiment:

Can the authors describe in detail how the strain was measured? Was extensometer used for the Gleeble testing. What was the gauge length of the specimen, which was used to calculate the strains and strain rates? It may be helpful to describe more about the Gleeble (brand and model) used for the study in the experiment section. It might also be good to add some information of the as-received material, e.g. initial (austenitic) microstructure.

3. Results and discussion:

Figure 3: Scattering of the data is more significant for the higher strain rates. In fact, with the linear fitting, many of the data fall away from the fitted curve. This deviation may explain the significant discrepancy between experiment and model calculation, for example that in figure 12. The authors also stated that J-C model has a better accuracy at low strain rates. The authors should discuss this deviation at high strain rates, resulting in poor accuracy in the predictions. The same issue applies to figures 6.

Page 6, Line 171: It is stated “J-C model ignores the influence of strain, strain rate and temperature. And it does not consider other factors”. The statements are not correct as the model actually involves strain rate and temperature terms. The authors perhaps meant the coupling effects as they stated in the conclusions section. Please revise the text and specify “other factors”.

Some minor comments:

The paper contains typographical errors. The authors should examine the paper to reduce the errors as much as possible.

Page 1, Line 28: The term “FEM” has not been defined. Perhaps change it to “finite element modeling (FEM)”.

Page 1, Line 34-35: it is stated “… strong chemical stability…”. The meaning of “steel with strong chemical stability” is unclear. Please remove or correct the text.

Page 3, Line 110: “…sample was heated to 1453K…” ® “…sample was heated to the target temperature…”

Page 4, Line 140: liner ® linear

Tables 2 and 3 need to be better aligned.

Page 17, Line 347: F.inite ® Finite

Author Response

Dear reviewer and editor

Thank you for your letter and for the reviewers’ comments concerning our manuscript entitled “Comparative Study on Constitutive Models for 21-4N Heat Resistant Steel During High Temperature Deformation” (ID: materials-518931). Those comments are all valuable and very helpful for revising and improving our paper, as well as the important guiding significance to our researches. We have studied comments carefully and have made correction which we hope meet with approval. Revised portion are marked in red in the paper.

The main corrections in the paper and the responds to the reviewer’s comments are as following:

Reviewers' comments:
Reviewer #1: Comparative Study on Constitutive Models for 21-4N Heat Resistant Steel During High Temperature Deformation
The paper presents interesting results and calculation data with respect to the high temperature deformation behavior of a heat resistant steel. It is admirable that the authors compared four different models considering strain rate- and temperature-dependent mechanical behavior. The results presented may be useful to the relevant community. I however believe that there are a few parts where the authors can improve the technical quality of the manuscript. I would like to comment on some points in the manuscript:

1. Experiment: Can the authors describe in detail how the strain was measured? Was extensometer used for the Gleeble testing. What was the gauge length of the specimen, which was used to calculate the strains and strain rates? It may be helpful to describe more about the Gleeble (brand and model) used for the study in the experiment section. It might also be good to add some information of the as-received material, e.g. initial (austenitic) microstructure.

Response: Thanks for the reviewer’s suggestion.

We accept your suggestion and have modified it in the paper.

The test was carried out on Gleeble-1500D high-temperature simulation test machine. The deformation temperature was measured by thermocouple, and the stress-strain data were collected through the computer control system.

2. Results and discussion: Figure 3: Scattering of the data is more significant for the higher strain rates. In fact, with the linear fitting, many of the data fall away from the fitted curve. This deviation may explain the significant discrepancy between experiment and model calculation, for example that in figure 12. The authors also stated that J-C model has a better accuracy at low strain rates. The authors should discuss this deviation at high strain rates, resulting in poor accuracy in the predictions. The same issue applies to figures 6.

Response: Thanks for the reviewer’s suggestion.

We accept your suggestion and have modified it in the paper.

When the deformation temperature is constant, the prediction accuracy of J-C model gradually decreases with the increase of strain rate. When the strain rate reaches 10s-1, the model basically loses its prediction ability. When the strain rate is 10s-1, as the deformation temperature increases, the accuracy of the model becomes lower and lower, and the prediction ability is lost when the deformation temperature is 1333K.

In addition, in this study, the accuracy of the J-C model only reaches the highest under the reference deformation condition. With the change of the deformation condition, the accuracy of the model also decreases, and the increase of data scattering is consistent with this. This is also the case in other literature (1. A comparative study on JohnsonCook, modified JohnsonCook and Arrhenius-type constitutive models to predict the high temperature flow stress in 20CrMo alloy steel. 2. Prediction on hot deformation behavior of spray-formed 7055 aluminum alloy via phenomenological models.).

3. Page 6, Line 171: It is stated “J-C model ignores the influence of strain, strain rate and temperature. And it does not consider other factors”. The statements are not correct as the model actually involves strain rate and temperature terms. The authors perhaps meant the coupling effects as they stated in the conclusions section. Please revise the text and specify “other factors”.

Response: Thanks for the reviewer’s suggestion.

We accept your suggestion and have modified it in the paper.

“Obviously, J-C model is established under the assumption that strain, strain rate and deformation temperature do not affect each other.”

4. Some minor comments: – The paper contains typographical errors. The authors should examine the paper to reduce the errors as much as possible.

Response: Thanks for the reviewer’s suggestion.

We accept your suggestion and have modified it in the paper.

5. Page 1, Line 28: The term “FEM” has not been defined. Perhaps change it to “finite element modeling (FEM)”.

Response: Thanks for the reviewer’s suggestion.

We accept your suggestion and have modified it in the paper.

Finally, a Finite Element Method (FEM) of the isothermal compression experiment for four models were established.

6. Page 1, Line 34-35: it is stated “… strong chemical stability…”. The meaning of “steel with strong chemical stability” is unclear. Please remove or correct the text.

Response: Thanks for the reviewer’s suggestion.

We accept your suggestion and have modified it in the paper.

21-4N is an austenitic heat-resistant steel with good mechanical properties at high temperature.

7. Page 3, Line 110: “…sample was heated to 1453K…” ® “…sample was heated to the target temperature…”

Response: Thanks for the reviewer’s suggestion.

We accept your suggestion and have modified it in the paper.

The sample was heated to the target temperature.

8. Page 4, Line 140: liner ® linear.

Response: Thanks for the reviewer’s suggestion.

We accept your suggestion and have modified it in the paper.

From Eq. (4),  is linear with .

9.
Tables 2 and 3 need to be better aligned.

Response: Thanks for the reviewer’s suggestion.

We accept your suggestion and have modified it in the paper.

Table.2 The values of  under different .

0.01

0.1

1

10

-0.00673

-0.00528

-0.00420

-0.00389

Table.3 Parameter values of 21-4N under different strains.

Parameter

ε

0.05

0.1

0.15

0.2

0.25

0.3

0.35

0.4

m

9.571

8.144

7.505

7.059

6.670

6.288

6.137

5.852

β

0.0665

0.0545

0.0495

0.0465

0.0441

0.0425

0.0417

0.0415

α

0.00694

0.00669

0.00660

0.00659

0.00660

0.00676

0.00679

0.00708

n

6.819

5.797

5.356

5.025

4.752

4.475

4.355

4.188

n1

12547.3

12957.7

12549.5

12985.7

13055.1

13255.2

13463.4

12806.4

Q

711389

624550

558860

542547

515813

493191

487504

445933

lnC

59.087

52.144

46.428

45.129

42.896

40.948

40.548

36.900

Parameter

ε

0.45

0.5

0.55

0.6

0.65

0.7

0.75

0.8

m

5.577

5.404

5.321

5.238

5.187

5.112

5.068

5.046

β

0.0405

0.0404

0.0406

0.0409

0.0414

0.0418

0.0423

0.0432

α

0.00726

0.00748

0.00763

0.00780

0.00798

0.00818

0.00835

0.00856

n

3.988

3.876

3.822

3.775

3.744

3.697

3.677

3.682

n1

13504.1

13429.8

13384.9

13221.8

13063.5

12831.4

12541.5

12421.2

Q

447772

432802

425345

414995

406660

394420

383423

380262

lnC

37.188

35.935

35.350

34.497

33.799

32.752

31.831

31.691

10. Page 17, Line 347: F.inite ® Finite.
Response: Thanks for the reviewer’s suggestion.

We accept your suggestion and have modified it in the paper.

Finite element simulation of four models.

We have tried our best to revise and improve the manuscript and made great changes in the manuscript according to the Reviwers′good comments. And here we list the changes. We also marked in red in revised paper. We appreciate for Editors/Reviewers’ warm work earnestly, and hope that the corrections will meet with approval. Once again, thank you very much for your comments and suggestions.

We look forward to your information about my revised papers and thank you for your good comments.

Yours sincerely,

Hongchao Ji

Reviewer 2 Report

An interesting article about significant cognitive problems. The introduction is sufficiently large but not carried out accurately. The Author analyzed and cited only/mostly Asian researchers. We can also find other researchers working in this field. For example:

-          W. Grzesik, P. Nieslony, P. Laskowski: Determination of Material Constitutive Laws for Inconel 718 Superalloy Under Different Strain Rates and Working Temperatures, J. Materials Engineering and Performance, 2017, Volume 26, Issue 12, pp 5705–5714

-          Özel, Tuğrul & Karpat, Yigit. (2007). Identification of Constitutive Material Model Parameters for High-Strain Rate Metal Cutting Conditions Using Evolutionary Computational Algorithms. Materials and Manufacturing Processes

The methodology for identifying the parameters of constitutive equations was presented very clearly. This is done well, but in such an article does not have to be done so thoroughly. I could not find precise information about the measurement method of the isothermal compression test. On what machine, under which conditions, how were technically implemented the changes of strain rate? That should be supplemented.

What is the reason for use reference strain rate as 0.01 1/s? The equations 28 and 29 presented the AARE and R. There is no explanation for the symbols used in them.

What was wanted to achieve through FEM tests? It has been confirmed that the constitutive models can be entered into the FEM program. But how have the results of the simulation to the real test? Could not you rate this? Also, the simulation results can be faster and correctly analyzed when the color bars rates have the same range for all tests. I think it can be easily corrected.

The conclusions are very short described. They only refer to conformity assessment of constitutive models with the results of stress tests. There is no conclusions to the evaluation of FEM simulation results. It is worth adding. Generally, an interesting and well-written article. Research into constitutive models is an important aspect of FEM analysis.

Author Response

Reviewer #2: 1. An interesting article about significant cognitive problems. The introduction is sufficiently large but not carried out accurately. The Author analyzed and cited only/mostly Asian researchers. We can also find other researchers working in this field. For example:

W. Grzesik, P. Nieslony, P. Laskowski: Determination of Material Constitutive Laws for Inconel 718 Superalloy Under Different Strain Rates and Working Temperatures, J. Materials Engineering and Performance, 2017, Volume 26, Issue 12, pp 5705–5714.

Özel, Tuğrul & Karpat, Yigit. (2007). Identification of Constitutive Material Model Parameters for High-Strain Rate Metal Cutting Conditions Using Evolutionary Computational Algorithms. Materials and Manufacturing Processes.

Response: Thanks for the reviewer’s suggestion.

We accept your suggestion and have modified it in the paper. And we have added two references:

Grzesik et al [17] established the J-C model of Inconel 718 nickel-based alloy, and conducted finite element simulation of turning and milling process of jet engine parts with MATLAB software, and obtained better prediction results. Tuğrul et al [18] proposed to use the evolutionary algorithm to determine the constitutive model parameters of metal materials under high strain rate cutting conditions, and compared with other methods, proved its excellent performance.

Bobbili et al [23] studied the modified J-C model, modified Khan-Huang-Liang (KHL) model and artificial neural network (ANN) model of Ti-13Nb-13Zr alloy respectively. Prawoto et al [24] studied the bi-ferritic martensite structure of two kinds of hypoeutectoid steel with different carbon content and alloyed elements, and proposed the failure criterion of J-C model to change the ferrite content. Ducobu et al [25] collected the parameters of the j-c model of Ti6Al4V under different deformation conditions from different literatures, and compared the prediction ability of the j-c model under different parameters, and obtained three parameter sets with better prediction performance. Ranc et al [26] used the j-c model to analyze the influence of thermal softening on energy. Shrot et al [27] proposed the Levenberg-Marquardt search algorithm to calculate parameterS of the J-C model, established the J-C model with a specific parameter, and used the model to predict the rheological behavior of materials.

[17] Grzesik W, Nieslony P, Laskowski P. Determination of Material Constitutive Laws for Inconel 718 Superalloy Under Different Strain Rates and Working Temperatures. Journal of Materials Engineering and Performance, 2017, 26(12):5705-5714.

[18] Tuğrul Ö, Yiğit K. Identification of Constitutive Material Model Parameters for High-Strain Rate Metal Cutting Conditions Using Evolutionary Computational Algorithms. Advanced Manufacturing Processes, 2007, 22(5):659-667.

[23] Bobbili R, Madhu V. Constitutive modeling and fracture behavior of a biomedical Ti-13Nb-13Zr alloy. Materials Science and Engineering: A, 2017, 700:82-91.

[24] Prawoto Y, Fanone M, Shahedi S, et al. Computational approach using Johnson-Cook model on dual phase steel. Computational Materials Science, 2012, 54:48-55.

[25] Ducobu F, Rivière-Lorphèvre, E, Filippi E. On the importance of the choice of the parameters of the Johnson-Cook constitutive model and their influence on the results of a Ti6Al4V orthogonal cutting model. International Journal of Mechanical Sciences, 2017, 122:143-155.

[26] Ranc N, Chrysochoos A. Calorimetric consequences of thermal softening in Johnson-Cook’s model. Mechanics of Materials, 2013, 65:44-55.

[27]S hrot A, BäKer M. Determination of Johnson-Cook parameters from machining simulations. Computational Materials Science, 2012, 52(1):298-304.

2.
The methodology for identifying the parameters of constitutive equations was presented very clearly. This is done well, but in such an article does not have to be done so thoroughly. I could not find precise information about the measurement method of the isothermal compression test. On what machine, under which conditions, how were technically implemented the changes of strain rate? That should be supplemented.

Response: Thanks for the reviewer’s suggestion.

We accept your suggestion and have modified it in the paper.

The test was carried out on Gleeble-1500D high-temperature simulation test machine. The deformation temperature was measured by thermocouple, and the stress-strain data were collected through the computer control system.

3. What is the reason for use reference strain rate as 0.01 1/s?

Response: Thanks for the reviewer’s suggestion. Here are some explanations:

The reference deformation temperature and strain rate were set at 1273K and 0.01s-1 respectively to facilitate the calculation process. The temperature and strain rate of quasi-static compression test can also be used as the reference deformation temperature and the reference strain rate. In addition, we also referred to some literatures (1. Zhao Y, Sun J, Li J, et al. A comparative study on Johnson-Cook and modified Johnson-Cook constitutive material model to predict the dynamic behavior laser additive manufacturing FeCr alloy[J]. Journal of Alloys and Compounds, 2017, 723:179-187. 2. An H, Xie G, Zhang H, et al. A comparative study on Johnson–Cook, modified Johnson–Cook and Arrhenius-type constitutive models to predict the high temperature flow stress in 20CrMo alloy steel[J]. Materials & Design, 2013, 52(24):677-685.) and found that they were set in the same way, indicating that it is feasible to use the lowest strain rate and the lowest deformation temperature within the range of experimental research as the reference strain rate and deformation temperature.

4. The equations 28 and 29 presented the AARE and R. There is no explanation for the symbols used in them.

Response: Thanks for the reviewer’s suggestion.

We accept your suggestion and have modified it in the paper. And we have marked references in the paper.

The average relative error (AARE) and correlation coefficient (R) is introduced to verify the accuracy of four models.

5. What was wanted to achieve through FEM tests? It has been confirmed that the constitutive models can be entered into the FEM program. But how have the results of the simulation to the real test? Could not you rate this? Also, the simulation results can be faster and correctly analyzed when the color bars rates have the same range for all tests. I think it can be easily corrected.

Response: Thanks for the reviewer’s suggestion.

We accept your suggestion and have modified it in the paper.

The main purpose of finite element simulation is to prove the reliability of the four models. The simulation results show that the model is correct and can be used for finite element simulation of material forming process.

Fig.18 The equivalent stress field, equivalent strain field and temperature field of Four models.

6. The conclusions are very short described. They only refer to conformity assessment of constitutive models with the results of stress tests. There is no conclusions to the evaluation of FEM simulation results. It is worth adding. Generally, an interesting and well-written article. Research into constitutive models is an important aspect of FEM analysis.

Response: Thanks for the reviewer’s suggestion.

We accept your suggestion and have modified it in the paper.

6. All the four models can be input into the finite element software for compression test simulation, and the simulation results are not much different from the experimental results, indicating that the four models established have certain practicability.

We have tried our best to revise and improve the manuscript and made great changes in the manuscript according to the Reviwers′good comments. And here we list the changes. We also marked in red in revised paper. We appreciate for Editors/Reviewers’ warm work earnestly, and hope that the corrections will meet with approval. Once again, thank you very much for your comments and suggestions.

We look forward to your information about my revised papers and thank you for your good comments.

Yours sincerely,

Hongchao Ji

Reviewer 3 Report

I have checked the aforementioned paper. I think that the paper is clearly introduced what they have done and the originality of their work. However, some modifications are required to clarify their work and understand clearly for the readers. Please see the following comments: The authors have conducted the variation of the experimental results with the numerical analysis using four models, but there are no clear information about the experiments how to examine the stress-strain characteristics. The authors have to describe the experimental conditions and setup together with the related results in details. Did you make compressive test? There are several (i) parameters on line 131 and 132 and (ii) some models selected, but the reasons for sections are not clearly described. Why did you use them? Please add the reasons for the determination. The authors have modified the model with e, e and T on line 171. Please tell us the significant of those parameters to obtain the high accuracy results. They have done the experiments at more than 1000K. At the high temperature heating, oxidation occurs in your sample. Is there any problem about this?

Author Response

Reviewer #3: I have checked the aforementioned paper. I think that the paper is clearly introduced what they have done and the originality of their work. However, some modifications are required to clarify their work and understand clearly for the readers. Please see the following comments:

1. The authors have conducted the variation of the experimental results with the numerical analysis using four models, but there are no clear information about the experiments how to examine the stress-strain characteristics. The authors have to describe the experimental conditions and setup together with the related results in details. Did you make compressive test?

Response: Thanks for the reviewer’s suggestion.

We accept your suggestion and have modified it in the paper.

The test was carried out on Gleeble-1500D high-temperature simulation test machine. The deformation temperature was measured by thermocouple, and the stress-strain data were collected through the computer control system.

Besides, we did compression tests, and here are the stress-strain curves:

2. There are several (i) parameters on line 131 and 132 and (ii) some models selected, but the reasons for sections are not clearly described. Why did you use them? Please add the reasons for the determination.

Response: Thanks for the reviewer’s suggestion. Here are some explanations:

The reference deformation temperature and strain rate were set at 1273K and 0.01s-1 respectively to facilitate the calculation process. The temperature and strain rate of quasi-static compression test can also be used as the reference deformation temperature and the reference strain rate. In addition, we also referred to some literatures (1. Zhao Y, Sun J, Li J, et al. A comparative study on Johnson-Cook and modified Johnson-Cook constitutive material model to predict the dynamic behavior laser additive manufacturing FeCr alloy[J]. Journal of Alloys and Compounds, 2017, 723:179-187. 2. An H, Xie G, Zhang H, et al. A comparative study on Johnson–Cook, modified Johnson–Cook and Arrhenius-type constitutive models to predict the high temperature flow stress in 20CrMo alloy steel[J]. Materials & Design, 2013, 52(24):677-685.) and found that they were set in the same way, indicating that it is feasible to use the lowest strain rate and the lowest deformation temperature within the range of experimental research as the reference strain rate and deformation temperature.

The yield stress of the material is the stress when the limit deviation of the linear stress-strain relationship reaches the specified value (usually the original standard distance of 0.2%). Therefore, this study takes the stress when the strain reaches 0.2% of the maximum deformation as the yield stress of the material under the reference deformation condition.

According to the relevant information, the melting point temperature of carbon steel is about 2500~2800. In this study, the melting point temperature is 2650, and the following formula is used to obtain the melting point temperature of carbon steel is 1727K.

3. The authors have modified the model with e, e and T on line 171. Please tell us the significant of those parameters to obtain the high accuracy results.

Response: Thanks for the reviewer’s suggestion.

We accept your suggestion and have modified it in the paper.

Obviously, J-C model is established under the assumption that strain (), strain rate () and deformation temperature (T) do not affect each other.

4. They have done the experiments at more than 1000K. At the high temperature heating, oxidation occurs in your sample. Is there any problem.

Response: Thanks for the reviewer’s suggestion. Here are some explanations:

21-4N heat resistant steel has good high temperature performance, suitable for long-term operation under high temperature conditions, and it has excellent oxidation resistance and stability. Besides, the time of experiment is very short, so the oxidation results can be ignored in the research.

We have tried our best to revise and improve the manuscript and made great changes in the manuscript according to the Reviwers′good comments. And here we list the changes. We also marked in red in revised paper. We appreciate for Editors/Reviewers’ warm work earnestly, and hope that the corrections will meet with approval. Once again, thank you very much for your comments and suggestions.

We look forward to your information about my revised papers and thank you for your good comments.

Yours sincerely,

Hongchao Ji

Round  2

Reviewer 1 Report

The authors have well incorporated their thoughts in the revised manuscript. The topics and calculation data will be useful for the community, and I recommend publication of this paper in Materials.